# Nutrigenetics-based intervention approach for adults with non-alcoholic fatty liver disease (NAFLD): study protocol for a randomised controlled feasibility trial

Laura Haigh [ORCID],[1,2] Stuart McPherson,[1,2] John C Mathers,[3] Quentin M Anstee[1,2]

JCM and QMA are joint senior authors.

[1]Regional Liver Unit, Newcastle upon Tyne Hospitals NHS Trust, Newcastle upon Tyne, UK
[2]Faculty of Medical Sciences, Newcastle University, Newcastle upon Tyne, UK
[3]Human Nutrition Research Centre, Newcastle University, Newcastle upon Tyne, UK

**Correspondence to**
Laura Haigh;
l.haigh3@newcastle.ac.uk

## ABSTRACT

**Introduction** Lifestyle interventions targeting weight loss and improved dietary patterns are the recommended treatment for non-alcoholic fatty liver disease (NAFLD). However, the effectiveness of current established diet therapies is suboptimal. The patatin-like phospholipase domain containing 3 (*PNPLA3*) gene modifies disease outcome and hepatic lipid handling, but the role of *PNPLA3* variants in modulating responsiveness to different diet therapies is unknown.

**Methods and analysis** This project aims to assess the feasibility of conducting a genotype-driven randomised controlled trial (RCT) investigating the differential response to a Mediterranean diet (MD) intervention of NAFLD patients according to genotype for the rs738409 (I148M) variant of *PNPLA3*. A single-centre randomised controlled feasibility trial will be undertaken. We will recruit 60 adults with NAFLD from a tertiary hepatology centre in England. In a cross-over design, participants will undertake Diet 1 (MD) and Diet 2 (control) for 4 weeks, in random order (1:1 allocation), separated by a 4 weeks washout period. Participants will complete one-to-one diet and lifestyle consultations at baseline, end of diet phase 1, end of washout and end of diet phase 2. Participants will be advised to maintain baseline levels of physical activity and body weight. The primary outcome is the acceptability and feasibility of the intervention protocol. Secondary outcomes include exploratory assessment of liver fibrosis biomarkers and lipid biomarkers.

**Ethics and dissemination** Ethical approval was granted by East of Scotland Research Ethics Service REC 1 (19/ES/0112). Results will be disseminated through peer-reviewed journals and presented at local, national and international meetings and conferences. The findings of this trial will lay the foundation for a future definitive RCT by informing trial design and optimising the intervention diets, instruments and procedures.

**Trial registration number** ISRCTN93410321.

## Strengths and limitations of this study

► The main strength of this trial is the use of explicit feasibility and acceptability criteria as primary outcomes.
► A unique feature of this trial is the use of metabolomics biomarkers to: (1) capture diet-mediated effects on lipid metabolism and (2) provide objective assessment of dietary intake.
► In the definitive randomised controlled trial that is expected to follow this feasibility trial, the use of a cross-over design will facilitate more precise comparisons between intervention/control diets on a within-participant basis.
► The effectiveness of washout to return outcome variables to baseline will be evaluated.
► The Mediterranean diet intervention has been systematically developed and informed by research evidence and patient and public feedback.
► This trial will generate qualitative and quantitative information to establish the feasibility and acceptability of the protocol.

plus inflammation, hepatocyte damage and progressive fibrosis (non-alcoholic steatohepatitis, NASH), that may potentially develop into cirrhosis and hepatocellular carcinoma.[3–5] NAFLD is strongly associated with obesity and cardiometabolic disease.[2 6]

NAFLD attributed rates of advanced liver disease and transplantation are increasing, but effective pharmacotherapy is unavailable.[2 3] A major challenge is the substantial interindividual variation in NAFLD susceptibility, progression and outcome, due partly to gene–environment interactions.[2 3] The patatin-like phospholipase domain containing 3 (*PNPLA3*) rs738409 single nucleotide polymorphism is a common modifier of disease outcome and its impact is amplified by adiposity.[7–10] This additive effect poses considerable concern, as patients with

## INTRODUCTION

Non-alcoholic fatty liver disease (NAFLD) affects 25% of the global population[1 2] and is a spectrum spanning steatosis (non-alcoholic fatty liver), through steatosis

NAFLD tend to consume excess calories as a consequence of poor-quality diets and sedentary lifestyles.[11–13]

The PNPLA3 protein has lipase activity and regulates lipid droplets in hepatocytes and hepatic stellate cells.[14] Several studies have contributed to understanding of its function in hepatic lipid handling.[15] Numerous genome-wide association studies have shown its association with the entire disease spectrum, which has been confirmed in various populations.[8 12 16 17] Therefore, among the identified NAFLD-related genes, *PNPLA3* has the most potential as a therapeutic target for NAFLD.[10 12]

The main treatment recommendations for NAFLD are lifestyle interventions targeting weight loss and improved dietary patterns.[4 5] The Mediterranean diet (MD) reduces steatosis, improves liver biochemistry and cardiometabolic dysfunction, with or without weight loss and is the most recommended dietary pattern in NAFLD.[5 18–23] There is strong and consistent evidence that the MD, has beneficial effects on NAFLD-associated conditions such as type 2 diabetes and the metabolic syndrome.[24] However, there is suboptimal response to current diet therapies in NAFLD and more effective approaches for enhancing adherence to diet therapies are needed.[11]

Personalised nutrition approaches that use information on individual participant characteristics to tailor the dietary intervention may improve the suboptimal response to current diet therapies.[25] Differences in gene sequence can alter the activity of encoded proteins and affect the response of individuals to dietary components.[26] Emergent research has shown nutritional regulation of *PNPLA3*[15 26] so that patients with NAFLD carrying the *PNPLA3* risk allele might benefit more from weight loss but less from omega-3 supplementation.[26–29] Diet lifestyle modification is more effective in decreasing liver steatosis in *PNPLA3 I148M* carriers than in non-carriers.[29 30] Liver fat content is influenced by the interaction between *PNPLA3* variants and high carbohydrate intake, specifically sugar.[31] A hypocaloric low-carbohydrate diet induced greater hepatic fat reduction in carriers homozygous for the rs738409 G allele in the *PNPLA3* gene compared with carriers of the rs738409 C allele, irrespective of weight loss.[32] However, the role of *PNPLA3* variants in influencing responsiveness to different diet therapies is poorly understood.

This project aims to determine whether it is feasible to conduct a randomised controlled trial (RCT) to investigate the impact of *PNPLA3* carriage on responsiveness to MD and NAFLD severity and to provide preliminary data to inform the development of a definitive RCT.

## METHODS AND ANALYSIS
### Study design and setting
This trial is a single-centre, randomised controlled feasibility trial. In a cross-over design, participants will be randomised to either Diet 1 (MD) or Diet 2 (control) for 4 weeks, in random order, separated by a 4 weeks washout period. All study visits will be conducted in the outpatient hepatology services, in The Newcastle upon Tyne Hospitals NHS Foundation Trust (NuTH), UK. An overview of the study design is provided in figure 1.

### Study objectives
The primary objective is to determine whether the protocol for a future definitive RCT is acceptable and feasible.
We will:
1. Determine the feasibility of recruitment and retention.
2. Determine the acceptability of the diets, instruments and procedures.
3. Evaluate adherence to, and completion of, the diets and procedures.
4. Evaluate implementation fidelity and how practicable it is to deliver the protocol in a clinical setting.
5. Estimate outcome variability.

Secondary objectives include collection of preliminary exploratory data on liver fibrosis biomarkers and lipid biomarkers, to determine if *PNPLA3* carriage influences

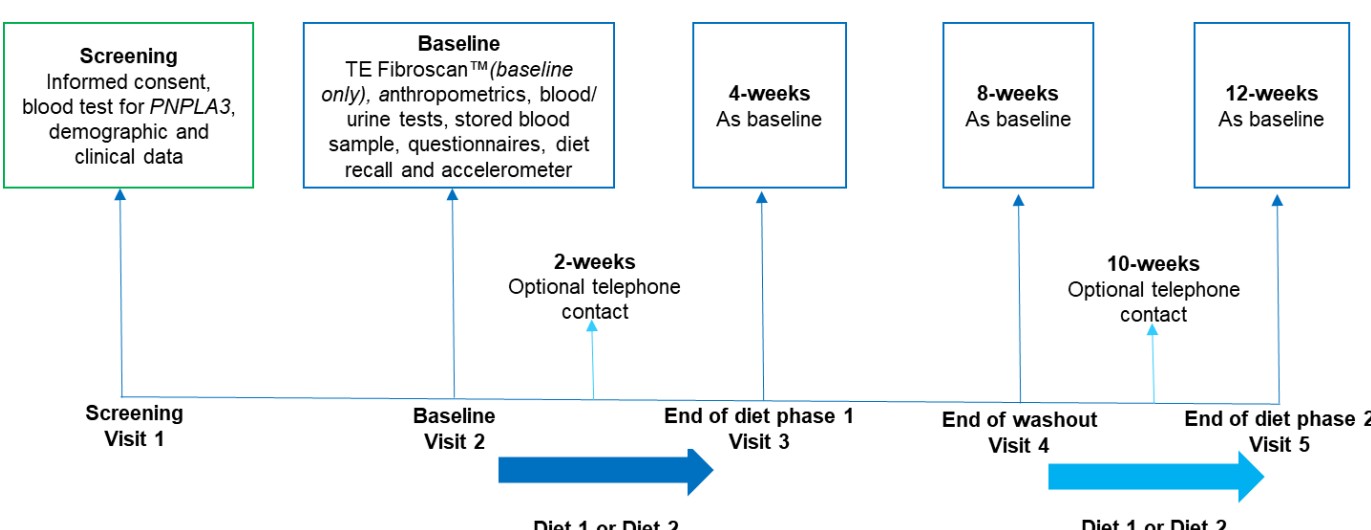

**Figure 1** Overview of the randomised controlled feasibility trial. PNPLA3, patatin-like phospholipase domain containing 3

## Box 1  Patient eligibility criteria

### Inclusion criteria
► 18–80 years old with NAFLD (confirmed on liver biopsy or by clinical diagnosis with imaging evidence of steatosis).
► Weekly alcohol consumption <14 (women)/<21 (men) units in the last 24 months.
► Weight stable (±5%) for previous 3 months.
► Capacity to provide informed consent.
► Ability to write and converse in English without assistance of an interpreter.

### Exclusion criteria
► All cancers within 5 years (except squamous cell carcinoma).
► Evidence of coexistent liver disease/presence of secondary causes of NAFLD (except Gilbert's syndrome).
► Decompensated NASH-cirrhosis (Child Pugh >6).
► Uncontrolled psychiatric disorder (eg, acute psychosis).
► Uncontrolled medical condition (eg, HbA1c >80 mmol/L or acute coronary event or stroke within 12 months).
► Active eating disorder.
► Active substance misuse.
► Other prescribed dietary regimens, food intolerances and/or food allergies.
► Mediterranean diet score (MEDAS) >8 (high MD consumption).
► Previous weight loss surgery.
► Taking antiobesity medications and/or engaged in structured, multicomponent weight management interventions (specialist, community or commercial providers).
► Insulin use.
► Pregnancy/lactation.

response to diet treatments. This mixed-methods feasibility trial will generate qualitative and quantitative information about feasibility and acceptability to inform a future definitive RCT.

### Sampling and recruitment
A recruitment target of 60 individuals with either imaging or histological evidence of NAFLD was established in accordance with published guidance.[33] This guidance suggests a sample of 30 individuals or greater, is sufficient to estimate a parameter with the necessary degree of precision.[33] Adults with NAFLD will be recruited from a tertiary hepatology centre in NuTH, which covers a population of approximately 3 million in northern UK. Potential participants meeting the inclusion and exclusion criteria will be identified from NuTH hepatology clinic lists and electronic records by members of the research and clinical care team. Potential participants will be approached during an appointment in liver outpatients or through screening of clinical notes and invited to participate in the study. The research team will contact patients after 48 hours to answer any queries. Enrolment will follow the receipt of full and written informed consent and successful screening. Recruitment and intervention delivery are anticipated to take place over 12 months.

### Inclusion and exclusion criteria
Participant eligibility criteria are detailed in box 1. We have chosen minimal exclusions to reflect the target population for a subsequent definitive RCT.

### Randomisation and allocation strategy
Eligible participants will be allocated in a 1:1 ratio to receive either Diet 1 or Diet 2 first using computer-generated randomisation. Prerandomisation genotyping for specific genetic variants (*PNPLA3 rs738409*) will be conducted using blood samples collected at screening. Interim analysis of the genotype distribution among the recruited cohort will be conducted at intervals as recruitment proceeds (eg, after one-third and two-thirds have been randomised); randomisation will then be stratified by *PNPLA3* status at baseline to ensure a balanced recruitment for *PNPLA3 rs738409* genetic status (wild-type, heterozygote, homozygote).

The major allele frequency for *PNPLA3 rs738409* is approximately 0.25 in the UK general population[34] and the minor allele frequency is greater in those with NAFLD.[17] Accordingly, we expect to reach the recruitment target for wild-type cases sooner than for carriers of the genetic variant and so some patients with a specific genotype may not be randomised if sufficient cases of a given genotype are already enroled. Those discharged from the study will be offered a one-to-one nutrition education and counselling consultation with a dietitian. It will not be possible to blind participants, clinicians or research investigators to which diet each participant is on in each study period.

### Trial procedures
The trial procedures are outlined in table 1. At an initial screening appointment, demographic, clinical, dietary and lifestyle data will be collected to determine patient eligibility and a blood sample will be taken for *PNPLA3 rs738409* genotyping. In addition, at baseline, liver steatosis will be assessed using controlled attenuation parameter measured contemporaneously with liver stiffness by Transient Elastography, Fibroscan.

Enroled participants will complete one-to-one diet and lifestyle consultations at baseline, end of diet phase 1 (4 weeks), end of washout (8 weeks) and end of diet phase 2 (12 weeks). At each of these timepoints, the following measures will be taken; anthropometry and body composition; blood biochemistry; dietary intake, urine samples and physical activity. The clinical status and medication consumption of participants will be checked at each timepoint. Patient-reported outcome data will be captured at the end of each diet phase (4 weeks and 12 weeks).

### Dietary intervention and control treatments
The experimental Diet 1 is a MD based on the traditional MD and MD pyramid.[35] The MD is characterised by a high quantity of plant-based foods, unrefined cereals, fruit and vegetables, olive oil and nuts; eating white meat, fish and legumes in moderation; restricting red and processed

**Table 1** Schedule of study measures

| Variable | Instrument | Screening | Baseline | End of diet phase 1 | End of washout | End of diet phase 2 |
|---|---|---|---|---|---|---|
| | | **Timepoint** | | | | |
| Inclusion/exclusion criteria | | × | | | | |
| Informed consent | | × | | | | |
| Demographic data and medical history | Self-report and clinical records | × | | | | |
| Genotyping | Venous blood sample | × | | | | |
| Anthropometrics: weight, height, waist and hip circumference | | | × | × | × | × |
| Whole body composition | Bioelectrical impedance analysis | | × | × | × | × |
| Cardio-metabolic measures: glucose, insulin, HbA1c, lipid profile and blood pressure | Fasting venous blood samples | | × | × | × | × |
| Liver function: LFTs, ferritin, FBC, CRP, lipid biomarkers | Fasting venous blood samples | | × | × | × | × |
| Liver steatosis: CAP | TE Fibroscan | | × | | | |
| Liver fibrosis: liver stiffness and PRO-C3 | TE Fibroscan | | × | | | |
| | Fasting venous blood samples | | × | × | × | × |
| Patient-reported outcomes | EQ5D, CLDQ-NASH and NASH-CHECK | | | × | | × |
| Physical activity | Accelerometer | | × | × | × | × |
| Dietary intake: dietary biomarkers, diet recall and MD questionnaire | Urine samples | | × | × | × | × |
| | INTAKE24 | | × | × | × | × |
| | MEDAS | | × | × | × | × |

CAP, controlled attenuation parameter; CLDQ-NASH, chronic liver disease questionnaire for non-alcoholic steatohepatitis; CRP, C-reactive protein; EQ-5D, euroqol five dimension scale; FBC, full blood count; HbA1c, glycated haemoglobin; LFTs, liver function tests; MD, Mediterranean diet; MEDAS, Mediterranean diet assessment score; PRO-C3, N-protease cleaved PIIINP neo-epitope; TE, transient elastography.

meats and sweets; and drinking red wine moderately.[36] The diet was designed to be easy to follow over 4 weeks, informed by research evidence[37 38] and the findings from our earlier pilot study with patients and the public that explored barriers and facilitators to adoption of a MD intervention.[39] They highlighted the importance of reducing the burden of dietary changes, diet supporters in the household, regular nutritional counselling and preference for face-to-face contacts. These ideas have been incorporated into the final design.

To reduce participant burden and facilitate changes in food environment, some intervention foods will be supplied as prepackaged ready meals (FreshPrepare) and extra virgin olive oil 750 mL (Filippo Berio). Ten prepackaged ready meals will be home delivered weekly, taken as two main meals per day (lunch and evening meal) for 5 days. The meals can be ordered online, and examples of the available options are detailed in online supplemental appendix table 1. The provision of these intervention foods will assist in standardising food consumption and minimise variability in dietary intake.

To enhance MD adoption, nutrition counselling and education will be provided one-to-one to participants during visits, by a dietitian. This will include advice on the selection and preparation of appropriate meals, snacks and drinks. The consultation incorporates the 'model and process for nutrition and dietetic practice',[40] and involves participant discussion. Personalisation of the diet for specific personal and sociocultural preferences will be provided.

A patient information booklet has been designed to explain NAFLD, the principles of MD and how it can be successfully followed. This will be given to participants as evidence-based written material from a credible source. Behaviour change techniques will be utilised to increase intervention effectiveness. These include barrier identification, problem solving, goal setting and action planning; social support; and instruction on how to perform a behaviour and behavioural practice.[41–43]

Diet 2 (control) will involve counselling participants to consume their habitual diet. During washout participants will be asked to return to their habitual diet. Participants will be asked to maintain baseline levels of physical activity and body weight (±3%) throughout the trial duration. The dietitian will contact participants by telephone midway through each diet phase to review progress, provide additional counselling and answer any queries.

### Outcome assessment and process evaluation
The following criteria will be assessed.

#### The feasibility of recruitment and retention
The consent rate (the number of eligible participants who consent divided by the total number who are eligible and who are invited to participate), the recruitment rate (the number of participants recruited per month) and the retention rate (the number of participants who complete follow-up data collection divided by the total number randomised) will be assessed using a trial log between baseline and end of study.

#### The acceptability of the diets, instruments and procedures
Patient-reported outcome (PRO) data will be used to identify impact of diet treatments and trial procedures on quality of life using the PROs 'EQ5D', 'CLDQ-NASH' and 'NASH-CHECK'.[44–46] At 12 weeks, open-response questions will be used to capture participant perceptions of: (1) the randomisation procedure, (2) the acceptability of the MD intervention and (3) the components of the measurement protocol. These data will be recorded (online supplemental appendix 2), and thematic analyses performed.

#### Adherence to, and completion of, the diets and procedures
Participant adherence to trial procedures will be assessed by tracking the number of completed visits, and the completeness of data collection will be assessed using the trial log between baseline and 12 weeks.

Diet adherence will be assessed in self-report measures; Mediterranean diet assessment score (MEDAS) is based on a small number of foods measured in servings/day or servings/week. Scores range between 0 and 14, and can be categorised as low, moderate or high consumption (<5, 6–9 and >10 points, respectively).[47] Dietary intake will be quantified using INTAKE24, an open-source computerised dietary recall system based on multiple-pass 24-hour recall.[48 49] INTAKE24 data will be converted to MD scores and Dietary Inflammatory Index will be calculated.[50]

In addition, dietary biomarkers will be quantified in urine to provide objective measures of dietary intake, without self-report bias.[51] Three spot urine samples will be collected at the beginning and end of each diet phase on non-consecutive days, including 1 weekend day, to provide estimates of habitual dietary intake. The three urine samples from each timepoint will be pooled and analysed using Ultra High-Performance Liquid Chromatography.[52]

#### Implementation fidelity and how practicable protocol processes are to deliver in a clinical setting
Data collection burden will be measured as the time taken to administer protocol processes. Participant processing time is the number of days from initial contact to enrolment. Both will be assessed using the trial log between baseline and end of study. To assess integrity and fidelity, a trial protocol checklist will be monitored with missing, incomplete or unreliable data recorded between baseline and end of study.

#### Preliminary exploratory data on liver fibrosis biomarkers and lipid biomarkers
In collaboration with partners in the EU IMI2-funded LITMUS biomarkers consortium (https://litmus-project.eu/), plasma will be analysed to characterise changes in the lipidome and metabolome to help elucidate mechanisms underlying specific changes in lipid metabolism. Plasma PRO-C3 concentration, as a marker of fibrogenesis, will be measured using competitive ELISAs.[53]

The data from this feasibility trial will be reported as descriptive summaries. Sample characteristics including age, gender, ethnicity, medical history and disease subclassification and severity will be presented. Although the trial is not powered to detect significant changes in clinical and lifestyle outcomes, key variables of interest (outlined in schedule of assessments) will be monitored and any preliminary changes reported. These data will determine the feasibility of testing components and enable the statistical power calculations for a subsequent RCT. The balance of the groups after randomisation will be explored. Descriptive statistics (mean, SD and counts (percentage) will summarise continuous and categorical data as appropriate. No imputation of missing data will be undertaken.

### Success criteria
The feasibility indicators are binary (successful or unsuccessful). 'Successful' would indicate the protocol is sufficiently robust to advance to a fully powered definitive RCT, while 'unsuccessful' indicates that protocol changes are required.[54]
1. The acceptability of diets, instruments and procedures.
2. Consent rate (25% of individuals consenting).
3. Recruitment rate (target 60 individuals/9 months, acceptable 45 individuals/9 months).
4. Retention rate (target 45 individuals/3 months, acceptable 36 individuals/3 months).
5. Participant adherence (75% completed visits/data collection).
6. Data collection burden (target >75% of individuals ≤1.5 hours, acceptable >75% of individuals <2 hours).
7. Participant processing time (mean time <14 days between initial contact to enrolment).
8. Trial protocol administration (<10% deviation from checklist).

## Patient and public involvement

The early stages of the research process involving the preparation of the research proposal were supported by a national liver patient support group (LIVErNORTH). LIVErNORTH collaborate on the joint production of the patient information booklet, which is used in one-to-one diet and lifestyle consultations. To enhance the development of the MD intervention the findings from an earlier pilot study[39] and research evidence were discussed with a patient panel (APEX) as well as patients attending clinical services in NuTH. APEX advised on the patient experience and assessed the burden of the trial instruments and procedures. This patient and public feedback was integrated into the final design. There are plans in the future to involve patients and the public as the trial progresses, guiding the dissemination of trial results to participants and relevant wider patient communities.

## DISCUSSION

There is an urgent need to identify potential therapeutic interventions that can prevent NAFLD progression and induce regression.[11] Lifestyle interventions to induce weight loss and improve dietary lifestyle patterns are the mainstay of NAFLD treatment.[4 5] However, diet lifestyle targets are often difficult to achieve in practice and definitive data are needed on the optimal strategies to enhance adherence to diet treatments.[11] Stratified and targeted diet therapies may be an advance on the relatively ineffective 'one-size fits all' treatments. To that end, there is a need to explore the underlying mechanisms through which genotype influences responses to dietary components, and the subsequent effects on NAFLD.

The primary objective of this trial is to determine the acceptability and feasibility of a nutrigenetic approach for adults with NAFLD. This trial adopts a mixed-methods approach designed to establish whether the protocol is sufficiently robust and to inform any necessary refinements to diets, instruments and procedures. In addition to the primary outcomes, the trial will provide data on variability of secondary outcomes for power calculations to inform the design of a subsequent RCT.

A unique feature of this trial is the use of metabolomics biomarkers to: (1) capture diet-mediated effects on lipid metabolism and (2) provide objective assessment of dietary intake. Metabolomics approaches provide novel insights into complex disease traits and provide a powerful tool to predict and monitor responsiveness to diet treatments for NAFLD.[55] The application of 'omics' could support the design of innovative and effective diet therapies across the full NAFLD spectrum, and to understand their mechanisms.[55] The use of self-reported dietary intake assessment methods presents significant challenges both in research and clinical practice[56] and some of these challenges can be overcome using urine-based biomarker approaches.[51] Thus, we will combine self-reported data with the quantification of urinary dietary biomarkers to overcome this weakness.

The main strength of this trial is the use of explicit feasibility and acceptability criteria as primary outcomes. Patient-reported outcome measures and open-response questions will be especially important to identify impact of diet treatments and trial procedures on quality of life and individuals' perceptions of trial participation. In addition, the use of a cross-over design will facilitate more precise comparisons between intervention/control diets on a within-participant basis.[57] This design is favoured in short-term trials of long-term conditions with intermediate outcomes.[57] Nevertheless, a limitation of this design is the possibility of carryover effects from one experimental period to the next. We have addressed this issue by including a 4-week washout period. The required duration of washout period is influenced by the nature and duration of the intervention but, in nutritional cross-over studies, 2–4 weeks is often sufficient.[58 59] This is confirmed by studies of metabolomics biomarkers in urine which show that these respond rapidly to dietary change.[60] Importantly, we will evaluate the effectiveness of washout to return outcome variables to baseline. Additionally, participants will be supplied with intervention foods and counselled to return to habitual dietary intake during washout. Advice will be given to maintain baseline levels of physical activity and body weight (±3%) throughout. The intervention period (4 weeks) is relatively short and is unlikely to reveal the full effects of the dietary intervention on markers of liver health. Previous research has found that brief and short-term interventions effectively modified MD adoption and maintenance up to 12 months in non-Mediterranean countries[39 61–63] and that, with appropriate support, dietary change can be sustained for up to 4 years in Spain.[64] However, this study is designed, primarily, to provide information on the acceptability and feasibility of the study protocol. Finally, the MD intervention in this trial has been systematically developed and informed by research evidence and patient and public feedback.

This trial will assess the feasibility of conducting a genotype-driven RCT investigating the differential response to a MD intervention of patients according to genotype for the rs738409 (I148M) variant of *PNPLA3*. The findings of this trial will lay the foundation for a future definitive RCT by informing trial design and optimising the intervention diets, instruments and procedures. In the longer run, the outcomes of this research programme may lead to fewer patients with NAFLD who need intensified medical treatment, reducing cost burden, and premature mortality and morbidity.

## Ethics and dissemination

Ethical approval was granted by East of Scotland Research Ethics Service REC 1 (19/ES/0112). NHS Research and Development approval was granted by the Newcastle upon Tyne Hospitals NHS Foundation Trust (R&D8985). This trial will be conducted to a high standard in accordance with the protocol, the principles of good clinical practice, relevant regulations, guidelines and with regard

to patient safety and welfare. The findings of this trial will be submitted for publication in peer-reviewed journals and presented at local, national and international meetings and conferences. A lay summary of results will be available for all study participants.

## Trial status

The first participant was enroled on 11 February 2020 with recruitment expected to be completed by 31 March 2021. The recruitment period has been extended to mitigate the potential impact of COVID-19 on the research plans.

**Contributors** LH, JCM and QMA conceived of the study. LH, SMcP, JCM and QMA contributed to design of the study. All authors revised the manuscript for important intellectual content with JCM and QMA giving final approval of the version to be published.

**Funding** This research was funded by the NIHR Newcastle Biomedical Research Centre (BRC) (grant number BRC-1215–20001). The NIHR Newcastle Biomedical Research Centre (BRC) is a partnership between Newcastle Hospitals NHS Foundation Trust and Newcastle University, funded by the National Institute for Health Research (NIHR). QMA is a member of the LITMUS (Liver Investigation: Testing Marker Utility in Steatohepatitis) consortium funded by the Innovative Medicines Initiative (IMI2) Programme of the European Union (Grant Agreement 777377).

**Disclaimer** The views expressed are those of the authors and not necessarily those of the NIHR or the Department of Health and Social Care.

**Competing interests** SMcP: Consultancy/speakers fees—Abbvie, Allergan, BMS, Gilead, Intercept, MSD, Novartis and Sequana. QMA: Research Grant Funding Abbvie, Allergan/Tobira, AstraZeneca, GlaxoSmithKline, Glympse Bio, Novartis Pharma AG, Pfizer Ltd, Vertex. Active Research Collaborations (including research collaborations supported through the EU IMI2 LITMUS Consortium*) Abbvie, Antaros Medical*, Allergan/Tobira*, AstraZeneca*, BMS*, Boehringer Ingelheim International GMBH*, Echosens*, Ellegaard Gottingen Minipigs AS*, Eli Lilly & Company Ltd*, Exalenz Bioscience Ltd*, Genfit SA*, Glympse Bio, GlaxoSmithKline, HistoIndex*, Intercept Pharma Europe Ltd*, iXscient Ltd*, Nordic Bioscience*, Novartis Pharma AG*, Novo Nordisk A/S*, One Way Liver Genomics SL*, Perspectum Diagnostics*, Pfizer Ltd*, Resoundant*, Sanofi-Aventis Deutschland GMBH*, SomaLogic Inc*, Takeda Pharmaceuticals International SA*. Consultancy 89Bio, Abbott Laboratories, Acuitas Medical, Allergan/Tobira, Altimmune, AstraZeneca, Axcella, Blade, BMS, BNN Cardio, Celgene, Cirius, CymaBay, EcoR1, E3Bio, Eli Lilly & Company Ltd, Galmed, Genentech, Genfit SA, Gilead, Grunthal, HistoIndex, Indalo, Imperial Innovations, Intercept Pharma Europe Ltd, Inventiva, IQVIA, Janssen, Madrigal, MedImmune, Metacrine, NewGene, NGMBio, North Sea Therapeutics, Novartis, Novo Nordisk A/S, Pfizer Ltd, Poxel, ProSciento, Raptor Pharma, Servier, Terns, Viking Therapeutics. Speaker Abbott Laboratories, Allergan/Tobira, BMS, Clinical Care Options, Falk, Fishawack, Genfit SA, Gilead, Integritas Communications, Kenes, MedScape. Royalties Elsevier Ltd (Davidson's Principles & Practice of Medicine textbook).

**Patient consent for publication** Not required.

**Provenance and peer review** Not commissioned; externally peer reviewed.

**ORCID iD**
Laura Haigh http://orcid.org/0000-0002-9229-4127

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
