## [Reviewer comments · BMJ Open]

ARTICLE DETAILS

TITLE (PROVISIONAL)	A Nutrigenetics-based Intervention Approach for Adults with Non-Alcoholic Fatty Liver Disease (NAFLD): Study Protocol for a Randomised Controlled Feasibility Trial
AUTHORS	Haigh, Laura; McPherson, Stuart; Mathers, John; Anstee, Quentin

VERSION 1 – REVIEW

REVIEWER	Zalilah Mohd Shariff Universiti Putra Malaysia
REVIEW RETURNED	08-Nov-2020

GENERAL COMMENTS	Reviewer's report – Manuscript BMJOPEN-2020-045922 The trial is well-thought of, and the manuscript is well-written and easy to read. The comments / suggestions provided below are for clarification of the points / statements indicated in the manuscript: 1. Suggested title: A Nutrigenetics-based Intervention Approach for Adults with Non-Alcoholic Fatty Liver Disease (NAFLD): Study Protocol for a Randomised Controlled Feasibility Trial2. Introduction (page 3) – Beside omega-3 supplementation conferring positive effect on NAFLD patients with PNPLA3 variants, what other diet treatments (similar or not similar to MD) that have been reported to show effects / no effects on these patients and the hypothesized mechanisms for response towards the diet treatments?3. Sampling and recruitment (page 4) – Is 60 a fixed sample size as recommended by the reference or is it based on authors' sample size calculation that accounts for effect size of the secondary outcomes? Provide a brief explanation to how 60 is derived even though it is based on the reference4. Inclusion and exclusion criteria (page 5) – the age range for patients to be recruited into the study is 18-80 years. Is there any rationale for the use of this wide age group? Although the effect of age can be controlled in the analysis, would it be better to exclude the elderly / older age groups i.e > 60 years?5. Dietary interventions (page 6) – in future RCT, will patients be provided with ready-made meals or will they be only advised to incorporate MD into their habitual diets? If the ready-made meals are not provided, how does the provision of read-made meals in this feasibility trial be relevant to the future RCT?6. Acceptability of diets/instruments/procedures (page 8) – are the perceptions of randomization procedure and components of instruments from trial patients or researchers? If patients, what aspects of these two require inputs from them? Is the qualitative inquiry going to be an individual in-depth interview or focus group discussion?
---

	7. Adherence to and completion of diets / procedures (page 8) – For the foods supplied to the patients in the trial, is there any intention to ask patients to provide photos of food waste to support acceptance / adherence to the provided foods or MD? 8. Analysis (page 9) – For the secondary outcomes, would authors consider examining these outcomes by the 3 categories of PNPLA3 (even though not necessarily to check for significance difference) as to align to authors' suggestions that this is a genotype-drive RCT that investigate differential response to MD intervention?
--	---

REVIEWER	Ian Rowe University of Leeds
REVIEW RETURNED	30-Nov-2020

GENERAL COMMENTS	This is a protocol to an interesting feasibility trial of a "nutrigenetic" intervention. The trial has received ethical approval and is underway. The manuscript describing the trial would benefit from some additional information regarding some specific points.  1. Background information. The manuscript would benefit from a more detailed description of the evidence that supports the importance of the PNPLA3 polymorphism on the development of NAFLD and specifically on the impact of dietary / lifestyle intervention according to the presence of that polymorphism. 2. Intervention selection. There are several potential dietary / lifestyle interventions that are recommended for weight loss in persons at risk of NAFLD. It is not clear why the mediterranean diet was selected for this study as opposed to other available interventions and this should be explained. 3. Genotyping and blinding. It is plausible that knowledge of the patient's genotype might influence patient and clinician behaviour during the trial. In the latter phases of the trial it seems inevitable that this information will be known by the investigators as recruitment will likely be limited to those patients with less frequent genotypes. Will patients be blinded to this information? And if so, when will they be informed? 4. Prior diet and lifestyle intervention. This patient group with NAFLD will often have received previous dietary / lifestyle advice, particularly as recruitment will be from a specialist centre with a longstanding interest in the management of NAFLD. I appreciate that the inclusion criteria are broad, but will prior dietary advice history be considered in recruitment? 5. Duration of intervention and outcomes considered for the main trial. I understand that the purpose of this trial is to determine the feasibility of a full trial. However, what are the projected outcomes for the full trial? Is a 4-week intervention likely for that study? If, as I think likely, the full trial intervention duration is longer are there existing data that supports 4-week intervention acceptability being maintained for a longer period? Considering the outcomes for the full trial, it is not clear to me what this would be and how they would be assessed. A number of measures are included though according to Table 2 it seems that liver fat assessed by the CAP is only being measured at baseline. This may have been a missed opportunity to assess short-term repeatability of the CAP in the context of such a clinical trial. Since the manuscript states that the
--

	information gained would be used to inform a power calculation for the full trial it would be interesting to understand how that information would actually be used. 6. PRO measurement. According to Table 2, it appears that PROs are not being measured at baseline and it is not clear to me why that would be the case. Justification of this should be included in the text.
--	--

REVIEWER	Anj Reddy La Trobe University, Australia
REVIEW RETURNED	21-Dec-2020

GENERAL COMMENTS	Dear Author(s), Congratulations on your work in conceptualising this study, I look forward to reading the results of your trial. Please see a few minor comments below: Table 1. Patient eligibility criteria - Time frame for NAFLD Diagnosis (i.e., biopsy or US within 6mo?) Page 6, line 1-2. Will dietitians/clinicians/research investigators be blinded to genetic status of individuals? This is unclear and may cause potential bias Page 7, lines 16-18. Body weight maintenance comment is somewhat misleading. If participants experience body composition changes (change in VF) whilst adhering to the MD and perceive this as weight loss, this advice may encourage them to eat more or discontinue habitual exercise. Query as to whether this should be advised to participants re body weight. Perhaps dietitian should monitor these changes and intervene where necessary. Page 8, line 21-22. "This mixed methods feasibility trial will generate qualitative and quantitative information about feasibility and acceptability to inform a future definitive RCT", this statement of mixed methods feasibility study should be placed much earlier in the paper. Page 8, line 32-33. Reference to 'follow up data' at baseline and 12-months – unclear what this is. The intervention length is 12 weeks, but 12 months is mentioned as follow up. What outcomes are being followed up? Assessment of dietary intake - 24 hr recall being used at each timepoint? Have authors considered using 3- or 7- day food diaries are much more comprehensive and provide a more accurate measure of habitual dietary intake. Inconsistency between urinary biomarkers of dietary adherence and diet recalls – 24 hr diet recalls to be collected at appointments 0-, 4-week, 8-week, 12-week whereas "three spot urine samples will be collected at baseline and after each diet treatment on non-consecutive days, including one weekend day". At the end of the MD will be collecting hydroxytyrosol? Which measure will reflect adherence to the control diet? Could authors be more explicit here please.
---

	Page 11, lines 21-24, as stated by authors this intervention length is short. There have been MedDiet interventions implemented in non-Med populations that have reported high adherence to diet and acceptability of diet. Therefore, should these comments be more specific, i.e., is this population in the UK the first of its kind. A 4-week intervention, though regular for feasibility studies, is short due to the issue of dietary adherence being sustainability of diet/prolonged adherence. Short term adherence has been shown to improve IR, lipids and IHL in Australian populations. Page 11, line 30-31, The “genotype driven” RCT approach is not clear throughout this protocol paper. Participants will be genotyped at screening and randomised based on their genotype, however personalised diet advice will not be provided based on genotype? Therefore, the “Nutrigenomics-based RCT” (as written in title) is somewhat misleading. The analysis of participant results and response to diet will be the main ‘nutrigenomics’ approach. Therefore, perhaps dietary responsiveness should be discussed within the discussion and perhaps authors could elude to the methodology of this analysis. Kind regards, AR
--	---

VERSION 1 – AUTHOR RESPONSE

Reviewer: 1

Prof. ZALILAH MOHD SHARIFF, University Putra Malaysia

Comments to the Author:

Reviewer’s report – Manuscript BMJOPEN-2020-045922 The trial is well-thought of, and the manuscript is well-written and easy to read. The comments / suggestions provided below are for clarification of the points / statements indicated in the manuscript:

Author response: Thank you, we appreciate your feedback.

1. Suggested title: A Nutrigenetics-based Intervention Approach for Adults with Non-Alcoholic Fatty Liver Disease (NAFLD): Study Protocol for a Randomised Controlled Feasibility Trial

Author response: Thank you for this suggestion; we have amended the title accordingly.

2. Introduction (page 3) – Beside omega-3 supplementation conferring positive effect on NAFLD patients with PNPLA3 variants, what other diet treatments (similar or not similar to MD) that have been reported to show effects / no effects on these patients and the hypothesized mechanisms for response towards the diet treatments?

Author response: We have expanded the Introduction (page 4, paragraph 1, lines 159-163) to include a summary of the emergent evidence on diet-gene interactions in NAFLD.

3. Sampling and recruitment (page 4) – Is 60 a fixed sample size as recommended by the reference or is it based on authors' sample size calculation that accounts for effect size of the secondary outcomes? Provide a brief explanation to how 60 is derived even though it is based on the reference.

Author response: Thank you for your comment. The sample size was derived from the reference paper, which suggests that a sample size of 30 patients or greater is sufficient to estimate a parameter with the necessary degree of precision. We have updated the manuscript text in (page 5, paragraph 1 lines 196-197).

4. Inclusion and exclusion criteria (page 5) – the age range for patients to be recruited into the study is 18-80 years. Is there any rationale for the use of this wide age group? Although the effect of age can be controlled in the analysis, would it be better to exclude the elderly / older age groups i.e > 60 years? Author response: Thank you for these helpful comments. We have not yet finalised the age range for a future definitive trial and so the outcomes of this feasibility trial, in particular the ages of participants recruited, will inform that aspect of inclusion criteria for the future study. At this stage, we see no clear rationale for excluding patients aged > 60 years.

5. Dietary interventions (page 6) – in future RCT, will patients be provided with ready-made meals or will they be only advised to incorporate MD into their habitual diets? If the ready-made meals are not provided, how does the provision of read-made meals in this feasibility trial be relevant to the future RCT?

Author response: Thank you for these important questions. We plan to provide freshly prepared ready meals to participants in the future definitive RCT so testing that aspect of the study feasibility is essential. In the later definitive study, we may make modifications to the meals offered, website design, and meal ordering processes based on the feedback provided in the current feasibility study.

6. Acceptability of diets/instruments/procedures (page 8) – are the perceptions of randomization procedure and components of instruments from trial patients or researchers? If patients, what aspects of these two require inputs from them? Is the qualitative inquiry going to be an individual in-depth interview or focus group discussion?

Author response: Thank you for your comments. We are using a series of open-response questions to capture participants perceptions of the randomisation procedure and instruments and, indeed, other aspects of the study protocol. In the Appendix, we have added a copy of the guide that is being used by investigators at the end of the trial. We have updated the manuscript text (page 9, paragraph 1, line 301).

7. Adherence to and completion of diets / procedures (page 8) – For the foods supplied to the patients in the trial, is there any intention to ask patients to provide photos of food waste to support acceptance / adherence to the provided foods or MD?

Author response: Thank you for this excellent suggestion., The current feasibility trial assesses adherence to the experimental diet in several ways including: by recording the number of meals consumed weekly, through use of INTAKE24 to assess total dietary intake, by use of the MEDAS screener to assess adherence to the Mediterranean dietary patterns and through use of urinary biomarkers of food intake. Diet acceptability is also captured using the guide detailed in response to Point 6, above.

8. Analysis (page 9) – For the secondary outcomes, would authors consider examining these outcomes by the 3 categories of PNPLA3 (even though not necessarily to check for significance difference) as to align to authors' suggestions that this is a genotype-drive RCT that investigate differential response to MD intervention?

Author response: Thank you. Yes, this is planned already.

Reviewer: 2

Dr. Ian Rowe, University of Leeds

Comments to the Author:

This is a protocol to an interesting feasibility trial of a "nutrigenetic" intervention. The trial has received ethical approval and is underway. The manuscript describing the trial would benefit from some additional information regarding some specific points.

Author response: Thank you.

1. Background information. The manuscript would benefit from a more detailed description of the evidence that supports the importance of the PNPLA3 polymorphism on the development of NAFLD and specifically on the impact of dietary / lifestyle intervention according to the presence of that polymorphism.

Author response: Thank you for your suggestion. Please see the response to Reviewer 1 point 2.

2. Intervention selection. There are several potential dietary / lifestyle interventions that are recommended for weight loss in persons at risk of NAFLD. It is not clear why the Mediterranean diet was selected for this study as opposed to other available interventions and this should be explained.

Author response: Thank you for your request for clarification. This feasibility trial is not designed to achieve weight loss. Instead, the intervention is designed to improve dietary composition. In the Introduction (page 3, paragraph 4, lines 148-151) we have summarised the main evidence for the Mediterranean diet (MD) in NAFLD. In the revised manuscript, we have strengthened that evidence by reference to the European Association for the Study of the Liver (EASL) clinical practice guidelines, which recommends the MD (page 3, paragraph 4, lines 148-151).

3. Genotyping and blinding. It is plausible that knowledge of the patient's genotype might influence patient and clinician behaviour during the trial. In the latter phases of the trial, it seems inevitable that this information will be known by the investigators as recruitment will likely be limited to those patients with less frequent genotypes. Will patients be blinded to this information? And if so, when will they be informed?

Author response: Thank you for these important questions. Participants are blinded to genotype information throughout the trial to minimise risk of influence on behaviour. When the trial is complete. Genotyping is being undertaken by a research assistant who has no direct role in running the intervention trial or in collecting outcome data. We will minimise potential influence on researcher behaviour by limiting access to genotypic information solely to the person responsible checking eligibility of participants to enter the trial.

4. Prior diet and lifestyle intervention. This patient group with NAFLD will often have received previous dietary / lifestyle advice, particularly as recruitment will be from a specialist centre with a longstanding interest in the management of NAFLD. I appreciate that the inclusion criteria are broad, but will prior dietary advice history be considered in recruitment?

Author response: Thank you for this point. Participants are excluded if they are currently engaged with another dietary regimen or dietary lifestyle service; already follow the MD; take Orlistat; or have undergone previous weight loss surgery. We consider that for potential transferability to the real-world clinic population, it is very useful to include a broad range of participants with and without prior dieting history.

5. Duration of intervention and outcomes considered for the main trial. I understand that the purpose of this trial is to determine the feasibility of a full trial. However, what are the projected outcomes for the full trial? Is a 4-week intervention likely for that study? If, as I think likely, the full trial intervention duration is longer are there existing data that supports 4-week intervention acceptability being maintained for a longer period? Considering the outcomes for the full trial, it is not clear to me what this would be and how they would be assessed.

Author response: We appreciate your feedback. You are correct in that this feasibility study does not involve piloting all aspects of a future RCT and, in particular, we are not attempting to assess long term diet acceptability or to assess efficacy. Further feasibility or pilot work may be necessary to inform the design of a later definitive trial. We have expanded the Discussion (page 11, paragraph 4, lines 405-408) to include evidence for increasing MD adherence in relevant populations for up to 4 years. A decision on the duration of a future definitive RCT will be based on the available evidence for kinetics of change in the chosen primary outcome which may be liver fat.

6. A number of measures are included though according to Table 2 it seems that liver fat assessed by the CAP is only being measured at baseline. This may have been a missed opportunity to assess short-term repeatability of the CAP in the context of such a clinical trial. Since the manuscript states that the information gained would be used to inform a power calculation for the full trial it would be interesting to understand how that information would actually be used.

Author response: Thank you for pointing this out. Although we agree that measurement of liver fat at baseline and end of intervention may be used in a future definitive RCT, we do not consider that making both measurements is essential for the current feasibility study. As noted in our answer to Point 5 above, the decision on the primary outcome of a future RCT will be informed by the wider evidence base and clinical experience and will be based on clinically meaningful changes (in liver fat) and the effect size associated with longer term MD interventions.

7. PRO measurement. According to Table 2, it appears that PROs are not being measured at baseline and it is not clear to me why that would be the case. Justification of this should be included in the text.

Author response: Thank you for this suggestion. In the future definitive trial, the measurement of PROs would be a secondary outcome. The focus of the current study is to assess the feasibility of collecting the relevant samples and measurements. Consequently, the current study is not designed to emulate every aspect of the future trial. We are capturing participant perceptions of the trial instruments, including collection of body fluids, which will inform future trial design.

Reviewer: 3

Dr. Anjana Reddy, La Trobe University

Comments to the Author:

Dear Author(s),

Congratulations on your work in conceptualising this study, I look forward to reading the results of your trial. Please see a few minor comments below:

Author response: Thank you, we appreciate your encouraging feedback.

1. Table 1. Patient eligibility criteria

Time frame for NAFLD Diagnosis (i.e., biopsy or US within 6mo?)

Author response: Thank you, for this question. The study inclusion criteria: 18-80 years old with NAFLD (confirmed on liver biopsy or by clinical diagnosis with imaging evidence of steatosis).

2. Page 6, line 1-2. Will dietitians/clinicians/research investigators be blinded to genetic status of individuals? This is unclear and may cause potential bias.

Author response: Thank you for this question. Please see our response to Reviewer 2 point 3.

3. Page 7, lines 16-18. Body weight maintenance comment is somewhat misleading. If participants experience body composition changes (change in VF) whilst adhering to the MD and perceive this as weight loss, this advice may encourage them to eat more or discontinue habitual exercise. Query as to whether this should be advised to participants re body weight. Perhaps dietitian should monitor these changes and intervene where necessary.

Author response: We appreciate your comments about changes in body weight. In the current short-term feasibility study, the participants are advised to maintain their body weight and so this study will assess the feasibility of this approach. Informed by the findings of this feasibility study, a future RCT may utilise an adapted protocol with strategies for adjusting dietary intake to address significant potential changes in body weight.

4. Page 8, line 21-22. "This mixed methods feasibility trial will generate qualitative and quantitative information about feasibility and acceptability to inform a future definitive RCT", this statement of

mixed methods feasibility study should be placed much earlier in the paper.

Author response: We think this is an excellent suggestion and so have moved this sentence to (page 4, paragraph 6, lines 189-191).

5. Page 8, line 32-33. Reference to 'follow up data' at baseline and 12-months – unclear what this is. The intervention length is 12 weeks, but 12 months is mentioned as follow up. What outcomes are being followed up?

Author response: We apologise for this confusion. The whole study is open for 12 months i.e. we aim to recruit the participants, to complete the intervention and to collect all data and samples within a 12-month period. You are correct – for a given participant, the intervention period is 12 weeks. Therefore, to avoid confusion, we have revised the outcome assessment and process evaluation section throughout to include the words “between baseline and end of study” (page 8, paragraph 2, line 293 and page 9, paragraph 4, lines 323-325).

6. Assessment of dietary intake - 24 hr recall being used at each timepoint? Have authors considered using 3- or 7- day food diaries are much more comprehensive and provide a more accurate measure of habitual dietary intake.

Author response: Thank you for these helpful comments. In this feasibility trial, we are assessing dietary intake in a number of ways including recording the number of MD meals consumed weekly, using the web-based 24-hour recall instrument INTAKE24, using the MEDAS questionnaire and via urinary biomarkers of dietary intake (an objective measure of dietary intake, without self-report bias). We have provided references in the manuscript (page 9, paragraph 2, lines 306-317) for the utility of these approaches.

7. Inconsistency between urinary biomarkers of dietary adherence and diet recalls – 24 hr diet recalls being collected at appointments 0-, 4-week, 8-week, 12-week whereas “three spot urine samples will be collected at baseline and after each diet treatment on non-consecutive days, including one weekend day”. At the end of the MD will be collecting hydroxytyrosol? Which measure will reflect adherence to the control diet? Could authors be more explicit here please.

Author response: Thank you for the opportunity to provide greater clarity. The collection of urinary biomarkers occurs at the beginning and end of each diet phase, which is in line with the other measures of dietary intake/ adherence. This has been amended on (page 9, paragraph 3, lines 314-315). We are assessing participants' habitual diet (Diet2) with the same instruments as those used for MD (Diet1). We are not collecting hydroxytyrosol, but this may be within the scope of a future RCT.

8. Page 11, lines 21-24, as stated by authors this intervention length is short. There have been MedDiet interventions implemented in non-Med populations that have reported high adherence to diet and acceptability of diet. Therefore, should these comments be more specific, i.e., is this population in the UK the first of its kind. A 4-week intervention, though regular for feasibility studies, is short due to the issue of dietary adherence being sustainability of diet/prolonged adherence. Short term adherence has been shown to improve IR, lipids and IHL in Australian populations.

Author response: Thank you these are useful points. Please see our response to Reviewer 2 point 5.

9. Page 11, line 30-31, The “genotype driven” RCT approach is not clear throughout this protocol paper. Participants will be genotyped at screening and randomised based on their genotype, however personalised diet advice will not be provided based on genotype? Therefore, the “Nutrigenomics-based RCT” (as written in title) is somewhat misleading. The analysis of participant results and response to diet will be the main ‘nutrigenomics’ approach. Therefore, perhaps dietary responsiveness should be discussed within the discussion and perhaps authors could elude to the methodology of this analysis.

Author response: This is an important consideration, but the current feasibility study does not include the delivery of personalised dietary advice. However, the feasibility data that we collect is expected to

inform the development of future studies. We envisage investigation of potential diet: gene interactions within the planned definitive RCT but, at this stage, we have not taken a decision whether that RCT will involve personalised dietary advice based on genotype.

VERSION 2 – REVIEW

REVIEWER	Zalilah Mohd Shariff Universiti Putra Malaysia - Malaysia
REVIEW RETURNED	13-Feb-2021

GENERAL COMMENTS	Authors have addressed all comments / inquiries of reviewer sufficiently
--

REVIEWER	Ian Rowe University of Leeds
REVIEW RETURNED	26-Feb-2021

GENERAL COMMENTS	The protocol is sound. I look forward to seeing the results of the study.
---

REVIEWER	Anj Reddy La Trobe University, Australia
REVIEW RETURNED	25-Feb-2021

GENERAL COMMENTS	Authors have made the appropriate amendments to the manuscript. Comments: Amendment to title is much more suitable for the contents of the article.
---